# Estimation of a Spectral Correlation Function Using a Time-Smoothing Cyclic Periodogram and FFT Interpolation—2N-FFT Algorithm [note 1]

**DOI:** 10.3390/s23010215

**Published:** 2022-12-25

**Authors:** Timofey Shevgunov, Evgeny Efimov, Oksana Guschina

**Affiliations:** Moscow Aviation Institute, Volokolamskoe Shosse 4, 125993 Moscow, Russia

**Keywords:** cyclostationarity, cyclostationary random process, spectral correlation function, spectral correlation density, cyclic spectrum, spectral correlation analysis, spectrum estimation, fast Fourier transform (FFT)

## Abstract

This article addresses the problem of estimating the spectral correlation function (SCF), which provides quantitative characterization in the frequency domain of wide-sense cyclostationary properties of random processes which are considered to be the theoretical models of observed time series or discrete-time signals. The theoretical framework behind the SCF estimation is briefly reviewed so that an important difference between the width of the resolution cell in bifrequency plane and the step between the centers of neighboring cells is highlighted. The outline of the proposed double-number fast Fourier transform algorithm (2N-FFT) is described in the paper as a sequence of steps directly leading to a digital signal processing technique. The 2N-FFT algorithm is derived from the time-smoothing approach to cyclic periodogram estimation where the spectral interpolation based on doubling the FFT base is employed. This guarantees that no cyclic frequency is left out of the coverage grid so that at least one resolution element intersects it. A numerical simulation involving two processes, a harmonic amplitude modulated by stationary noise and a binary-pulse amplitude-modulated train, demonstrated that their cyclic frequencies are estimated with a high accuracy, reaching the size of step between resolution cells. In addition, the SCF components estimated by the proposed algorithm are shown to be similar to the curves provided by the theoretical models of the observed processes. The comparison between the proposed algorithm and the well-known FFT accumulation method in terms of computational complexity and required memory size reveals the cases where the 2N-FFT algorithm offers a reasonable trade-off.

## 1. Introduction

The signal processing methods exploited in the majority of modern telecommunication and radar systems are typically based on the assumption that signals under processing can be modeled as realizations of some random processes [1]. The simplest assumption about the property of those processes is their wide-sense stationarity (WSS). The relative simplicity of the way that WSS processes can be described in time and frequency domain determines its extreme popularity among researchers dealing with any sort of signal processing in the fields related to electrical and electronic science, and far beyond it. Furthermore, the general theoretical frameworks aiming at solving typical problems such as signal detection [2] and estimation of their parameters [3] have been basically established on the conjecture that signals of interest and interfering noise belong to either the class of WSS random processes or deterministic waveforms with possibly unknown parameters. However, a lot of research in signal processing conducted so far [4] and some that is going on [5] declares that there is inevitable loss in the information about important features of the natural and artificial processes producing many types of observable signals if the models chosen for their description are narrowed down to stationary ones. That information being extracted in a proper way could fruitfully be used for gaining an increase in the accuracy of various techniques developed for signal detection, parameter estimation, classification, source localization, etc.

Second-order cyclostationarity, or wide-sense cyclostationarity, appears to be the intrinsic property exhibited by processes generating signals belonging to different classes. Thus, there are plenty of examples where it was reasonably taken into account in many research fields and engineering applications, including communication signals with various modulation schemes [6], radio signals used in passive radars [7,8], electromagnetic measurements with near-field probes [9,10], spectral sensing in cognitive radio [11], mechanical vibration of rotary machines [12,13], radio astronomy [14,15], power analysis of electric circuits [16], generalized detection [17,18] under conditions of the least certainty, electromagnetic compatibility [19,20], detection of vital signs in radar response [21,22], drone navigation signals [23] and others.

The first function that is traditionally introduced for the quantitative characterization of the wide-sense cyclostationarity exhibiting by random process x(t) is its cyclic autocorrelation function (CACF) [1]. CACF can be defined as follows:(1)$$R_{x}^{\alpha }\left(\tau \right)=E\left\{\lim_{B\to \infty }\;\frac{1}{B}\int_{-B/2\;}^{B/2\;}x\left(t+\tau /2\;\right)x^{{}_{*}}\left(t-\tau /2\;\right)\exp\left(-j2\pi \alpha t\right)dt\right\},$$
where the superscript ∗ denotes the complex conjugation, E stands for the probabilistic expectation operator, α is a cyclic frequency, or a cycle, which is the key parameter of the transformation. The signal is eventually said to exhibit ordinary wide-sense cyclostationary properties if its CACF is nonzero for cyclic frequencies α taking their values out of some countable set A (capital Greek alpha letter). If this set consists of a single element, $A=\left\{0\right\}$, the process x(t) is typically called wide-sense stationary, which can be considered as a particular case of the more general class of cyclostationary processes. In addition, it is worth noting that set A is generally assumed not to have any finite accumulation points [24]. Otherwise, it will lead to models involving the concept of generalized cyclostatioraity [25], which stands beyond the scope of the current paper.

Each component $R_{x}^{\alpha }\left(\tau \right)$ contributing to CACF (Equation 1) is basically a function depending on the argument τ, which can be interpreted as the time shift while the parameter α remains fixed during the transformation. It allows one to apply the Fourier transform to τ in order to obtain the counterpart of CACF related to an appropriate frequency domain denoted by *f*. The result is known as the spectral correlation function (SCF) of the random process and can be formally introduced as:(2)$$S_{x}^{\alpha }\left(f\right)=\int_{-\infty }^{\infty }R_{x}^{\alpha }\left(\tau \right)\exp\left(-j2\pi f\tau \right)d\tau ,$$
where the integration is supposed to be made either in Riemann’s or in the generalized sense depending on what class of functions $R_{x}^{\alpha }\left(\tau \right)$ belongs to. Thus, if the CACF is not absolutely integrable, the attempt at integration in the sense of distribution [26] may still remain a preferable option.

Although the CACF can be considered as a set of functions $S_{x}^{\alpha }\left(f\right),\alpha \in A$, it is possible to introduce another bifrequency characteristic—spectral correlation density (SCD). The components of SCF relating to the cyclic frequencies taken all together can straightforwardly be used for writing SCD via explicit expression:(3)$$S_{x}\left(f,\alpha \right)=\sum_{\nu \in A}S_{x}^{\alpha }\left(f\right)\delta \left(\alpha -\nu \right).$$

As can be seen in (Equation 3), SCD is a bispectral function whose arguments are two continuous frequency variables, in contrast to SCF $S_{x}^{\alpha }\left(f\right)$, which depends on discrete parameter α. The term δ(α−ν) designating a generalized function in (Equation 3) actually defines the so-called delta-fence. Roughly speaking, it can be simply described or visualized as a distribution whose support is a straight line parallel to the *f*-axis, and its width is infinitesimal along the α-axis.

Although both CACF and SCF contain the same information about the process, since they match a signal-spectrum Fourier pair, SCF can be preferred to CACF in many cases of applied spectral analysis, since both its variables, α and *f*, share the same frequency domain. It allows performing the further processing, e.g., axis rotation in the estimates, significantly easier, which can be important for processing some types of signals, e.g., chirp-modulated ones [27,28]. In addition, the features revealing cyclostationary behavior of the signal appear to form patterns which are more concentrated in the frequency domain than in time domain.

The components of SCF, as well as CACF, can be derived analytically for the known theoretical models of random processes using the probabilistic approach, or ensemble averaging [29]; a thorough example of the theoretical derivation can also be found in [30]. On the other hand, the researchers who typically encounter digital signals in practical analysis may want to estimate SCF by processing only finite-length samples of a time-series under observation. An estimator of SCF will indeed evaluate the sample SCD as a function of frequency pair (f,α), which, being properly scaled, can be considered as an estimate of the model SCF [31]. At the first glance, the estimation of $S_{x}^{\alpha }\left(f\right)$ at a chosen point (f,α) in the bifrequency plane does not seem to be a difficult issue. It may simply be carried out in two steps. The first one consists of computing the values of $R_{x}^{\alpha }\left(\tau \right)$ at the points of an appropriate grid $\left\{\tau_{n}\right\}$ in the dimension of τ argument while the chosen value of α remains fixed. This can be done immediately utilizing (Equation 1), where the limiting and expectation operators are both omitted. The second step is evaluating the Fourier-transform integral (Equation 2) at the chosen point in frequency *f* domain using those values of SCF $R_{x}^{\alpha }\left(\tau_{n}\right)$.

In many practical cases, a researcher will be interested in estimating SCF at many points simultaneously. Such an observation of SCF in a wide area of the bifrequency plane becomes especially crucial if the actual values of α depicting a signal are not known a priory, or if the cyclic frequency is one of the signal parameters to be estimated. The blunt approach where the SCF is estimated separately at many points has a serious drawback: extremely poor computational performance. Thus, the estimation of SCF in a wide range of frequency *f* and cyclic frequency α values remains an emerging topic in practical analysis of hidden periodicity in the signals which are assumed to be modeled as the realizations of cyclostationary random processes.

The estimators evaluating the SCF in a wide range of frequencies can generally be divided into two groups with respect to the fact whether the averaging is conducted in time or frequency domain [32]. It should not be a surprise that the greater attention has been focused on the estimators implementing the time-smoothing approach; they seem to be more valuable for at least three reasons. First, the time smoothing methods can be described as a generalization of the well-known Welch’s averaged periodogram method [33]. From this point of view, a typical SCF estimate will contain Welch’s periodogram as its natural part, which is the slice at zero cyclic frequency. That makes understanding of cyclic characteristics easier by expanding an analogy of the stationary characteristic to the cyclostationary ones. Second, the time smoothing methods allow processing the input samples at the rate they arrive at the input of the estimator, in contrast to the estimators realizing the frequency smoothing method. The latter require the full information about the estimated spectrum of the processed signal, which means a demand for all samples to be stored before the main processing starts. This feature of time-smoothing methods also allows designing a tracking cyclic estimator useful in the case of cyclic features changing slowly over a long observation. Finally, the computational process in the time-smoothing methods can be organized as a parallel or multithread execution in modern systems whose architecture may be based on multicore central processing units, graphic processing units or VLSI circuits [34].

One of the most widely known methods for cyclostationary characteristics estimation is the FFT accumulation method (FAM) [35], which was developed in the late 1980s. Their authors’ aim seemed to be the design of a memory-saving algorithm that would efficiently exploit the fast Fourier transform (FFT), which is rather a computationally effective method in digital signal processing. Despite the fact that the FAM is rather scalable and computationally effective, its possesses a disadvantage that makes its application unreliable in many cases. Thus, the SCF estimate computed by means of FAM cannot provide complete coverage of the bifrequency plane (α,f) due to the inevitable regular degradation of the resolution cells. That shrinkage results in almost the half of the bispectral plane turning out to be hardly tiled with the cells, providing sufficient width as a resolution element. Another algorithm proposed by the authors of FAM is the spectral strip correlation algorithm (SSCA) [36]. In contrast to FAM, SSCA covers the bispectral plane completely, but the underlying grid, i.e., the set of nodes where the resolution elements are evaluated, ceases to be rectangular. This makes further extraction of the slices along the frequency or cyclic frequency axis quite difficult, since it is no longer possible to take the appropriate row or column of the resulting rectangular matrix related to a requested frequency value. In order to obtain such slices, one would have to employ a complex interpolation technique that would in turn bring its own errors and negative performance effects. Another disadvantage of SSCA is that the averaged fragments of the input sequence can be separated by no more than one sampling period. That requirement significantly reduces the overall performance and increases the amount of memory required by the algorithm’s implementation compared to the methods where larger steps well retain the algorithm efficiency.

Another example of an out-of-the-box algorithm that implements a SCF estimator performing in wide frequency range can be found in [37], Chapter 5. The solution proposed there was developed into ready-to-use MATLAB code which invokes some standard functions delivered in its Signal Processing library. Those functions are conventionally used for estimating a complex-valued cross periodogram of two sampled signals but had been properly adapted for performing a cyclic correlation estimation according to its direct definition in the theory of spectral correlation [38].

The overview of the foundation lying in the basement of the majority of existing techniques leading to SCF estimators can be found in [31], where the common issues of choosing more suitable window and tapering functions as well as their influences on the sizes of the resolution cells are addressed. Further research of those authors in this field led to some reliable solutions [39], which are expected to be suitable in many practical cases. Finally, they came up with an implementation of a faster algorithm [40] where some sort of approximation is exploited.

An example of SCF estimator based on an alternative approach is given in [41], where such a two-dimensional window function that cannot be expressed as a product of two one-dimensional functions is applied to the outer product of two correlating signals. That led to the algorithm directly utilizing a two-dimensional Fourier transform [42], which determines its relatively high computational efficiency.

In spite of the fact that there exist SCF estimators performing in wide frequency ranges, we suppose that there is still room for yet another estimator if it is able to show a sufficient computational effectiveness while minimizing the drawbacks, such as the degradation of the resolution of the cells or an uneven grid of the nodes in the bifrequency plane where the estimates are evaluated. Although this algorithm was developed by the authors as early as in 2014, it remained unavailable for a wider community for several years, until it was reported at conference© 2021 IEEE [43]. Since the format of that conference paper did not let to provide all the details of the method, the current paper is aimed at eliminating this gap by giving a thorough description of the algorithm, accompanied by graphical illustrations explaining some of its essential operations. Compared to [43], this paper contains a completely new example involving a pulse-amplitude modulation (PAM) and a section containing a comparison of computational complexity between FAM and the proposed algorithm. All simulation examples are now provided with all necessary spectral and cyclic sample characteristics. The latter are also shown in figures with more close zooms, where the selective property of cyclic component is evident.

The rest of the paper is organized as follows. Section 2 narrates a brief overview of the cyclic periodogram time-smoothing approach to the estimation of SCF; the spectral resolution issue is pointed out in particular. In Section 3, the suggested double-number FFT (2N-FFT) algorithm is described in the form ready for immediate implementation as a signal-processing procedure. The simulation results are shown in Section 4, where the 2N-FFT algorithm was utilized for estimating SCF for the realizations of two signals which are famous for exhibiting strong cyclostationary behavior. The comparison between the proposed algorithm and FAM is given in Section 5. The paper ends with conclusions in Section 6.

## 2. Time-Smoothing SCF Estimator

The estimated spectral correlation function (SCF) can be generally considered to be a complex-valued function depending on two variables: the frequency *f* and the cyclic frequency α. The estimation of SCF based on time-smoothing approach [32] consists of processing continuous time signal x(t) of the finite-length *T* in this way:(4)$${\hat{S}}_{x}\left(f,\alpha \right)=\frac{1}{K}\sum_{k=0}^{K-1}X_{T_{W}}\left(t_{k},f+\frac{\alpha }{2}\right)X_{T_{W}}^{*}\left(t_{k},f-\frac{\alpha }{2}\right),$$
where *K* is the total number of generally overlapping fragments, $t_{k}$ denotes the starting point of the *k*th fragment and the current spectrum $X_{T_{W}}\left(t_{k},f\right)$ is obtained via Fourier transform of the signal fragments of the length $T_{W}$:(5)$$X_{T}\left(t_{k},f\right)=\frac{1}{\sqrt{T_{W}}}\int_{t_{k}}^{t_{k}+T_{W}}x\left(t\right)\exp\left(-j2\pi ft\right)dt.$$

In the vast majority of practical cases, the procedure of SCF estimation is assumed to be based on specific processing the input sequence of digital samples x[n] of the finite length *N* obtained by sampling the continuous-time signal x(t) with the sampling period $T_{s}$ within total observation time $T_{x}=NT_{s}$. The unique region of the principal support for the SCF of discrete-time signal x[n] is the area of the bispectral plane (f,α) within the following bounds:(6)$$\left|\alpha \right|+2\left|f\right|\leq F_{s},$$
where $F_{s}=1/T$ is the sampling frequency.

As is shown in Figure 1a, the region of support is a diamond, or rhombus, placed at the origin of the bifrequency plane. The main diagonals of the rhombus completely match the axes. They have different lengths. The length of the one along the frequency *f* axis is the sampling frequency $F_{s}$, and the length of the other one, which is along cyclic frequency α, turns out to be twice the size: $2F_{s}$. However, when it comes to processing the discrete-time signal $x\left[n\right]=x\left(nT_{s}\right)$, or time series, SCF has a particular structure beyond the principal rhombus. Thus, SCF turns into a function which is periodic with respect to both its arguments:(7)$${\hat{S}}_{\mathrm{DT}}\left(f,\alpha \right)=\hat{S}\left(f-\left(p-q\right)F_{s}/2\;,\;\alpha -\left(p+q\right)F_{s}\right)$$
where *p* and *q* are integers: $\left(p,q\right)\in Z^{2}$. This pair of indices can be thought as auxiliary axes *p* and *q*, respectively, which are shown in Figure 1a. That makes it easy to reveal how each diamond tile can uniquely be translated into the principal support at the origin, since each tile contains the copy of the same information about SCF but corresponds to its own unique index pair, (p,q).

In order to perform the averaging, *K* generally overlapping fragments of the length of *W* samples each, indexed by *k*, are extracted from the input time series x[n], 0≤k≤K−1:(8)$$x_{k}\left[n\right]=x\left[n+kR\right],\;0\leq n\leq W-1,$$
where *R* is the distance measured in samples between the starting points of two consequent fragments: 1≤R≤W, as is depicted in Figure 2. This fragmentation can be equivalently described as applying the rectangular window of length $T_{W}=WT_{s}$ moving over the input signal x(t) with the step *R*. After the signal is multiplied by the samples of the window in particular position, only those samples remain non-zero which happen to be covered by the non-zero elements of the window.

The important characteristics of any SCF estimator are the shape and size of its resolution cells [32,35]. The resolution cell describes the area which is integrally associated with the point estimates described by (f,α). The resolution cell of rectangular shape is shown in Figure 1b. The width of the resolution cell Δf alongside the frequency axis *f* is determined by the natural length of the fragment $x_{k}\left(t\right)$ or the length of its related sampled version $x_{k}\left[n\right]$ measured in samples:(9)$$\Delta f=\frac{1}{T_{W}}=\frac{F_{s}}{W}.$$

In contrast, the width of the resolution cell Δα alongside cyclic frequency axis α is the reciprocal of the total observation time $T_{x}$:(10)$$\Delta \alpha =\frac{1}{T_{x}}=\frac{F_{s}}{N}.$$

In order to avoid potential missing of any significant parts of SCF during its estimation, the whole area of the support region (Equation 6) in the bispectral plane should be fully covered with resolution cells. A simple yet reasonable scheme may consist of covering the support region of SCF by the resolution cells of the same width, regardless of their position in the plane. The cell can overlap some of its neighboring cells, but no gaps are allowed. As is shown in Figure 1b, the resolution cells generally turn out to be wide alongside the frequency axis *f*, whereas they are much narrower alongside the cyclic frequency axis α:(11)$$\frac{\Delta f}{\Delta \alpha }=\frac{T_{x}}{T}=\frac{N}{W}\gg 1.$$

A higher resolution in the dimension of cyclic frequency α allows one to solve more effectively a wide class of problems, including signal detection and signal selection, since the methods can be based on the cyclostationary features of the processed signals which do not appear in the domain of frequency *f*.

The practical digital signal processing leading to further SCF processing implies the estimation of its samples at the nodes of a rectangular grid of equidistantly distributed points. Each node of the grid corresponds to the center of the related resolution cell shown in Figure 1b. The steps alongside the dimensions of the frequency *f* and cyclic frequency α axes, denoted as δf and δα correspondingly, can generally be chosen as different. The evaluated matrix containing SCF samples can be therefore treated as a function depending on integer indices *l* and *m*. The relation between those sampled values and the continuous-time SCF estimation (Equation 4) can be established via a direct sampling formula:(12)$$\hat{S}\left[m,l\right]=\hat{S}\left(m\delta f,l\delta \alpha \right).$$

Since the full representation of the estimated SCF can be achieved within the region of its support defined by (Equation 6), it becomes sufficient to calculate only those samples of the SCF which are in the positions enumerated by pairs (l,m) satisfying the following condition:(13)$$\delta \alpha \left|l\right|+2\delta f\left|m\right|\leq F_{s}.$$

At least two reasonable suggestions should be taking into consideration in order to achieve higher performance of time-smoothing SCF estimator. The first consists of the usage of FFT for spectrum estimation whenever it is possible. The second is utilizing as few samples of the estimated spectra as possible to perform all the necessary operation to obtain the SCF estimate evaluated at a given point $\hat{S}\left[m,l\right]$.

## 3. 2N-FFT Algorithm

The quick look at the bispectral plane shown in Figure 1b could help one to come up with the basic idea that, in order to avoid gaps in the cyclic frequency α, the related step δα in covering grid should not exceed the width of the resolution cell Δα (Equation 10). Thus, let the step be chosen as the minimal value, which means it will be: δα=Δα. In order to calculate SCF estimation in each node of the grid, the step in the frequency *f* dimension must be equal exactly to the half δf=δα/2 for the following reason. The estimation at a given point ${\hat{S}}_{x}\left(f,\Delta \alpha \right)$ can be evaluated using two spectral samples as long as the values of the spectrum $X_{T_{W}}\left(f\right)$ at f±Δα/2 are known. In turn, that requires that grid of nodes is dense enough to allow the immediate multiplication without additional interpolation. Thus, the size of the step alongside axes *f* should be chosen as
(14)$$\delta f=\frac{\delta \alpha }{2}=\frac{\Delta \alpha }{2}=\frac{1}{2T_{x}}=\frac{F_{s}}{2N}.$$

The total number of points is the same 2N for both axes *f* and α despite the fact that thy cover different frequency ranges—they are $F_{s}$ and $2F_{s}$, respectively. The name 2N-FFT for this algorithm was proposed based on that fact.

The evaluation of the spectrum (Equation 2) was carried out by means of FFT. However, if FFT is taken on the input fragment $x_{k}\left[n\right]$ of length *W*, it will compute values of the spectrum only related to *W* frequency points that are multiples of $1/T_{W}=F_{s}/W$. Therefore, in order to obtain values over a more dense grid, at frequencies which are multiples of $\delta f=\Delta \alpha /2=1/(2T_{x})$, an interpolation procedure should be conducted. Here, we chose the well-known technique [44] carried out in spectrum interpolation using the same FFT algorithm only. It consists of enlarging the FFT base by padding the sequence $x_{k}\left[n\right]$ in time with 2N−W zero samples. In other words, it makes the total number of samples in each fragment equal to the desired length 2N:(15)$$\hat{x_{k}}\left[n\right]=\left\{\begin{array}{cc} x_{k}\left[n\right], & 0\leq n\leq W-1;\\ 0, & \mathrm{otherwise}. \end{array}\right.$$

Then FFT is applied to the augmented series yielding the spectrum samples of the *k*th fragment ${\overset{˜}{X}}_{k}\left(f\right)$ at points $m/(2T_{x})$ indexed by m=0,1,…,2N−1:(16)$${\overset{˜}{X}}_{k}\left(\frac{mF_{s}}{2N}\right)={\overset{˜}{X}}_{k}\left[m\right]=\frac{2N}{\sqrt{W}}\mathrm{FFT}\left\{{\overset{˜}{x}}_{k}\left[n\right]\right\}=\frac{1}{\sqrt{W}}\sum_{n=0}^{W-1}{\overset{˜}{x}}_{k}\left[n\right]e^{-j\frac{\pi mn}{N}},$$
where FFT stands for the forward version of the fast Fourier transform [45] with a 1/2N multiplier.

The above-described interpolation procedure allows getting the samples of the spectrum (Equation 2) distributed with even density well enough for evaluating the partial SCF of the *k*-th signal fragment. However, the effective width alongside frequency axis *f* of each resolution element shown in Figure 1b has not changed and remains $1/T_{W}$.

The spectrum of each fragment ${\overset{˜}{X}}_{k}\left[m\right]$ is computed as if the starting point of the fragment is zero. Therefore, the actual staring point of the *k*th fragment should be taken into account in order to provide the coherence between the spectra of the fragments. Since the latter is the key point of the SCF accumulation in (Equation 4), the phase correction has to be conducted. This important step consists of returning the starting time of the *k*-th signal fragment to the spectrum calculated using FFT directly as if this fragment started at t=0. The corrected spectrum ${\overset{˜}{X}}_{k}^{C}\left[m\right]$ is computed according to time shifting property in the frequency domain:(17)$${\overset{˜}{X}}_{k}^{C}\left[m\right]={\overset{˜}{X}}_{k}\left[m\right]\times \exp\left(-j\frac{\pi mkR}{N}\right).$$

The next step of the 2N-FFT algorithm is building up two square matrices denoted as **XR**(k) and XL(k). They both are of size 2N×2N. The first row of the XR(k) matrix matches the row vector of the spectrum (Equation 17). Then, each matrix row, starting from the second, is obtained via the circular shift in the previous row by one element to the right. The structure of this matrix is shown in Figure 3. The rows of the matrix XL(k) are calculated in the similar way but with the direction of the circular shifts reversed. It can be noticed that the matrices XR(k) and XL(k) have the specific structures known as cirulant and anticirculant matrices [46], respectively.

The final step of 2N-FFT consists of averaging the cross-spectra obtained for all the fragments, similarly to how it can be done during continuous time processing (Equation 4). Thus, the resultant matrix SM is calculated via averaging of the product of two matrices, XR(k) and XL(k):(18)$$\mathrm{SM}=\frac{1}{K}\sum_{k=0}^{K-1}\mathrm{XL}\left(k\right)\circ XR^{*}\left(k\right),$$
where ∘ stands for element-wise matrix multiplication, or Hadamard product, and ∗ applied to a matrix means the complex conjugation of its elements without the transposition of the matrix itself.

The computed elements in the resultant matrix SM are the SCF values estimated at the centers of resolution cells in accordance with the previously defined node grid with steps δf and δα:(19)$$\hat{S}\left(\frac{mF_{s}}{2N},\frac{lF_{s}}{N}\right)=\hat{S}\left[m,l\right]=SM_{l+1,m+1}.$$

The elements contained in matrix SM correspond to the estimated values of the SCF lying within the rectangle in the bifrequency plane: $\left(f,\alpha \right)\in \left[0,\;F_{s}\right)\times \left[0,\;2F_{s}\right)$. That area of the bispectral plane is shown in Figure 1a by the dashed rectangle. The diamond in its center corresponds to the index pair (p,q)=(1,0) according to (Equation 7). In order to find the principal support region, that is, the diamond-shaped region at (p,q)=(0,0), the periodic property (Equation 7) of the SCF is used. For instance, the center point $\hat{S}\left[0,0\right]$ translates to the point $\hat{S}\left[N,N\right]$ in the center of the estimated rectangle. This change is equivalent to the mapping the rectangle covered by the evaluated matrix to the centered one $\left(f,\alpha \right)\in \left[-F_{s}/2\;,\;F_{s}/2\;\right)\times \left[-F_{s},\;F_{s}\right)$. Actually, there is no transformation to be conducted on the elements of SM.

The final step that is aimed at improving the visualization of the SCF as a two-dimensional function is extracting the principal rhombus, which is shown centered at the origin in Figure 1a. In order to carve the diamond out of the averaged matrix SM, an appropriate mask ought to be applied:(20)$$SM_{⋄}=\mathrm{SM}\circ J,$$
where ${\mathrm{SM}}_{⋄}$ denotes the rhombus-shaped matrix containing meaningful elements in the diamond concentrated in its central part, and J is the mask matrix of size 2N×2N whose elements are defined as follows:(21)$$J_{l,m}=\left\{\begin{array}{cc} 1, & \left|l-N-1\right|+\left|m-N-1/2\right|<N;\\ \mathrm{NaN}, & \mathrm{otherwise}. \end{array}\right.$$

The NaN is acronym for “not a number”. This special value, predefined in many software systems and packages of applied mathematics, allows one to mark the variable for special purposes, e.g., as containing a missing value or a result of an uncertain mathematical operation, such as “0/0”. The advanced software will treat this value in a proper way, clearly distinguishing it from an ordinary zero value. For instance, NaN can be painted with a special color in an intensity diagram. If the software does not support NaN, it should simply be replaced with 0 in (Equation 21). However, this may require an additional attention to correct processing the values laying out of the principal rhombus.

## 4. Simulation Results

For the purpose of demonstrating the proposed interpolating 2N-FFT algorithm, two well-known examples [29] made of the signals exhibiting strong second-order stationarity were chosen.

The first example was an amplitude-modulated (AM) signal x(t), which was a sinusoidal-wave carrier modulated by a baseband wide-sense stationary Gaussian random process:(22)$$x\left(t\right)=\left[s(t)⋆h(t)\right]\cos\left(2\pi F_{0}t+\varphi_{0}\right),$$
where s(t) is the white Gaussian noise; h(t) is the impulse response of a low pass filter (LPF) whose cutoff frequency is $F_{\max}$; $F_{0}$ stands for the central frequency of the carrier; $\phi_{0}$ is the carrier’s initial phase, a nuisance parameter here; ⋆ denotes the linear convolution here. In the considered example, the carrier frequency was set to $F_{0}=6$ MHz, the LPF cutoff frequency $F_{\max}=1\;\mathrm{MHz}$, the carrier’s initial phase was random, the sampling period $T_{s}=44\;\mathrm{ns}$ and the total length of the sampled sequence to be processed was set as N=8192.

The typical realization of the process in the time domain is in Figure 4a, and the module of its sample spectral density is shown in The typical realization of the process in the time domain is in Figure 4b. The spectrum occupies the frequency band of 2 MHz full width centered at $F_{0}=6\;\mathrm{MHz}$ exhibiting the behavior of a typical wide-band radio signal in both time and frequency domains.

As long as the SCF is a complex-valued two-dimensional function dependent on (f,α), a «frequency–cyclic frequency» pair, a possible way to visualize it will be drawing a surface three-dimensional plot or colored diagram, where the color will show the absolute value of the function taken at the point. The cyclic features usually reveal there as thin lines oriented along the dimension of frequency *f* axis in the SCF plane [47]. Therefore, one may well rely on some criterion to distinguish it from the additive noise [48] or other components of the signal [49], which can produce a significant value at a local points due to the randomness of the estimate. The simple yet quite effective criterion can be based on the pseudo-power accumulated over the whole slice alongside frequency *f* axis, which was introduced in [50] and further applied in [51]:(23)$$\overset{˜}{P}\left(\alpha \right)=\sum_{m}\left|{\hat{S}}_{x}\left(\frac{mF_{s}}{2N},\frac{lF_{s}}{N}\right)\right|,$$
where the summation by *m* is carried out within the support region of the SCF demarked in (Equation 13). If zero cyclic frequency is considered, i.e., α=0, or l=0, this integral characteristic relates to the total power of the process, and power spectral density (PSD) $S_{x}^{0}\left(f\right)$ can be thought as the distribution of this power in the frequency *f* domain. In a similar manner, the scalars concentrated at other cyclic frequencies α≠0 can be treated as power-like quantities. In contrast, since the cyclic representation (Equation 1) does not provide the orthogonal decomposition of the signal x(t) itself, there is no sense in the alternative averaging over cyclic frequency dimensions for the fixed frequency *f*.

The absolute value of the SCF evaluated for the signal x(t) (Equation 22) and its corresponding integral characteristic $\overset{˜}{P}\left(\alpha \right)$ are shown in Figure 5, where the principal rhombus at the origin is shown with the mask (Equation 21) applied. Four wide stripes whose crossings generate four diamonds of the high intensity can be easily recognized in the figure. The centers of two of them lay down in the vicinity of $\pm F_{0}$ and on the line of zero cyclic frequency. Two others are at zero value of frequency *f* and in the vicinity of the cycles α equal to double carrier frequency $\pm 2F_{0}$. Those diamonds highlight the areas of where the spectral correlation takes relatively higher values. The zoomed-in part of the diagram is shown in Figure 6, although the exact value of the cycle responsible for the hidden periodic features is not clear. That is exactly the issue which is going to be solved with the integral characteristic (Equation 23). Its plot is shown attached on the right-hand side so as to to share the vertical axis α with the color two-dimensional diagram of the SCF. The narrow—literally from one-sample—spikes at $-2F_{0},\;0,\;2F_{0}$, sharply point out the cyclic frequencies exhibited by the AM signal which are expected to be zero and double the carrier frequencies according to theory the model [6].

Figure 7a shows the plot where integral characteristic alone indicates a clear peak at α near −12 MHz which is close to the model value $\alpha_{0}=-2F_{0}$. The error is less than 0.1%. Figure 7b shows the slice of the SCF at that frequency $S_{x}^{-2F_{0}}\left(f\right)$.

The pulse–amplitude modulation (PAM) signal y(t) has been chosen as the second example possessing strong cyclostationarity for the evaluating of the capability of the proposed 2N-FFT algorithm:(24)$$y\left(t\right)=\sum_{n=0}^{M-1}C_{n}\mathrm{rect}\left(\frac{t-nT}{\tau }\right),$$
where *T* denotes the period, τ is the width of each pulse, *M* is number of the pulses in the train, $\left\{C_{n}\right\}$ is a sequence of transmitted binary codes being modeled as a sequence of independent, identically distributed binary random variables taking values {−1,1} with the equal probability of 0.5. The waveform of a single pulse shaping the sequence is rectangular and can be formally written as
(25)$$\mathrm{rect}\left(v\right)=\left\{\begin{array}{cc} 1, & |v|<0.5;\\ 0.5, & |v|=0.5;\\ 0, & \mathrm{otherwise}. \end{array}\right.$$

For the numerical simulation, the pulse width was set to be equal to its period τ=T=704 ns. The sampling period: $T_{s}=$ 44 ns. The total length of the sequence: N= 8192. A typical realization of the process in the time domain is shown in Figure 8a, and the module of its sample spectrum density in the frequency domain is shown in Figure 8b.

The absolute value of the estimated SCF for the signal y(t) is shown in Figure 9, together with the its integral characteristic. The spectral correlation plot there is more complicated than the one exhibiting properties of the AM signal in Figure 5. Nevertheless, surrounding the origin there can be seen an area of stronger correlation, which is actually formed as the intersection of two vast stripes—their width is about 2 MHz, which is approximately the reciprocal of 1/τ. The integral characteristic will help again to disclose the set of cyclic frequencies α exhibiting the second-order periodicity. Those cycles are multiples of 1/T combined together with zero frequency. A zoomed-in version of the area of interest is shown in Figure 10, where the thin horizontal lines can also be clearly identified.

Those slices of the estimated SCF which correspond α=n/T,n∈Z may be visually compared to the SCF components provided by theoretical probabilistic model [1,30] with the assumption of the everlasting pulse train [1], or where the finite limits 1 and M in (Equation 24) are replaced by −∞ and +∞, respectively. Figure 11a shows the plot of the computed integral characteristic, where the peaks clearly stand out and are painted in different colors. Figure 11b shows theoretical and estimated curves drawn in dashed and solid lines so that the curves related to the same cyclic frequency are shown with the same color. The pair-wise comparison allows us to conclude that curves in the pairs are similar to each other, which proves the efficacy of the proposed estimation technique.

## 5. Algorithmic Complexity

The FFT accumulation method (FAM) is a well-known method that produces a large number of point-wise estimates of a spectral correlation function. It is used in this paper as a reference for the qualitative comparison of the algorithmic complexity. Therefore, the basic steps of FAM are briefly described in terms of computational complexity and memory costs. Afterwards, a similar analysis is presented for the algorithm proposed in this paper.

The basic steps in implementing the FAM algorithm in software typically consist of the following [52]:Sub-block matrix extraction is shaping the input signal of length *N* of complex values into a sub-block matrix of size W×P, where P=N/L, which is filled by sliding a window of size *W* with stride *L*. That operation will require memory allocation for PW complex values.Application of a data-tapering window consists of multiplying each column of the previously constructed matrix by a window function of the same length *W* that requires PW multiplications with no additional memory costs, since the old elements of the matrix may safely be discarded; thus, multiplication can be done in-place.The first Fourier transform is then applied to each of the matrix columns. That requires *P* times the FFT of size *W*; again, the previous data in the matrix may safely be discarded.Phase correction is required for taking into account the relative delays that are between the starting point related to each column in the matrix, since any information about the delays happens to be lost once FFT is applied to the column vectors. The corresponding coefficients are explicitly calculated, and then the correction requires PW multiplications.The second Fourier transform is then applied to the products of each matrix row with a complex conjugate of another row. That transformation is the key step of accumulation that reveals the hidden second-order periodicity intrinsically embedded in the signal. That operation is carried out $W^{2}$ times in total, and each time requires *W* multiplications to compute an element-wise row product. Then, a FFT of size *P* is applied to the product. Thus, there are in total $W^{3}$ multiplications and *W* rounds of FFT (size *P*), and a buffer of size $PW^{2}$ is to be reserved for storing the estimates.

As soon as the bispectral plane is considered to be completely covered with the estimates after the above-listed steps, the general result regarding FAM can be reached. The discussion from this point of view may be carried out in a dual manner: for the values of *W* and *L* being fixed and *N* varying, or for the values of *N* and *L* being fixed while *W* varies. Both cases are summarized in Table 1.

In contrast to how it is performed by the FAM, the proposed 2N-FFT approach seeks full bispectral coverage by explicitly widenin theg FFT base to 2N, which is expected to demand somewhat more computational resources. The basics steps in implementing 2N-FFT are the following:Allocating a buffer matrix of size 2N×2N is carried out for accumulation of the partial estimates, which requires $4N^{2}$ complex values. Here is the entrance to the main algorithm loop that processes P=N/L data chunks.Obtaining the *i*-th data chunk and weighting it with a window function: a memory buffer of *W* complex values is required accompanied by *W* multiplications, yielding a total of PW multiplications as soon as the whole loop is considered.The Fourier transform of the *i*-th chunk that within a loop requires *P* rounds of FFT of size 2N.The phase correction is necessary due to similar reasons for its being conducted for the FAM, so this step costs PW mutiplications.The shifts are performed over the the resultant vector to build the circulant and anti-circulant matrices. Then, the latter are conjugated, and the partial estimate is added to the accumulation buffer. Within a loop, that requires three times $4N^{2}$ complex values for matrices (if those values are not multiplied by *P* for the buffers in steps i−1, they may be reused in the *i*-th step) and *P* times $4N^{2}$ multiplications and additions.

The summary of computational costs described above is presented in Table 2.

The conducted comparison of both algorithms revealed that FAM is more computationally efficient if the values of *L* and *W* are fixed, whereas 2N-FFT offers better performance in regards to *W*. The total number of the points being estimated within the principal domain of the bispectral plane is $PW^{2}$ for FAM, and the value of *P* approaches N/L as *N* grows while the stride *L* remains the same. In contrast, 2N-FFT yields $4N^{2}$ estimates, that is, $4NL/W^{2}$ times more estimates than FAM. Since the often used [53] relation between the window size and stride is W=4L, the number of points provided by 2N-FFT is N/W times more than by FAM, which coincides with the ratio (Equation 11). In other words, 2N-FFT provides the characteristics implicitly interpolated among the frequency *f* dimension, so that δf=δα. Nevertheless, the necessary depth of the full coverage of the bispectral plane depends on resources available as well as on further processing stages where the estimates are involved, which leaves enough room for a trade-off based on the user’s choice.

## 6. Conclusions

The 2N-FFT algorithm presented in the paper further develops the basic concepts of the accumulation of sample cyclic periodogram in a time-averaging manner. The issue of insufficient resolution in the dimension of the cyclic frequency is overcome by extending the number of correlating points in the frequency domain. This is achieved with the simple yet effective interpolation technique consisting of the increasing in the base of FFT up to the double value of the total number of samples available in the processed time series. One of the benefits of the proposed estimator is the fact that its estimates cover the entire bispectral plane with the resolution cells of the same size uniformly placed at the nodes of the rectangular grid. This make the estimated samples of SCF reliable and convenient to further processing. The convenience here means that the estimates can be easily used for further steps of processing via matrix operations involving elementary row and column manipulations. The authors believe that the proposed algorithm may well be chosen as a clear reference for the verification of other faster SCF estimating algorithms. This algorithm can also serve an educational example providing deeper insight about the resolution problem specific to a cyclostationry signals subdomain.

The results of the numerical simulation performed with the examples of AM and PAM signals indicate that the curves provided by 2N-FFT algorithm merely align with the curves related to theoretical models of the simulated processes. Although the simulation was carried out in the absence of noise, a noise-like floor was present due to the intrinsic randomness of the process. It is also possible to notice some signs of the FFT leakage, where the neighboring cyclic frequencies are affected by the strong component at the neighboring cyclic frequencies.

One of the disadvantages of the 2N-FFT algorithm consists in its higher demand for computer memory in comparison with such algorithms as FAM or SSCA. This issue arises due to the necessity of processing complex-valued matrices of 2N×2N size in order to handle the full coverage of the SCF support region. The total amount of the memory required to store the data cannot be less than $8N^{2}$ floating-point cells of the chosen machine data type, presumably single or double precision. On the other hand, 2N-FFT has more steep memory consumption requirements for *N* values remaining fixed, which provides a sort of trade-off. The computational performance of 2N-FFT algorithm will also crucially depend on the general performance of the program library used in the backend of the chosen target platform for implementation of linear algebra operations.

The proposed method is promising for further development. The first direction may consist of seeking a windowing function weighing each of the signal fragments in order to decrease the leakage in both frequency and cyclic frequency dimensions. The second direction would be coping with the main disadvantage of the algorithm—finding the way to reduce the amount of memory required for storing the estimates. This can be carried out with introduction of decimation in the frequency domain because the width of the resolution cells along the frequency axis is considerably larger than the size of the step between two samples in that direction. Finally, the possible increase in the performance can be reasonably expected as soon as the parallel execution is introduced for estimating in different evaluation of the partial SCF estimates.

## Figures and Tables

**Figure 1 sensors-23-00215-f001:**
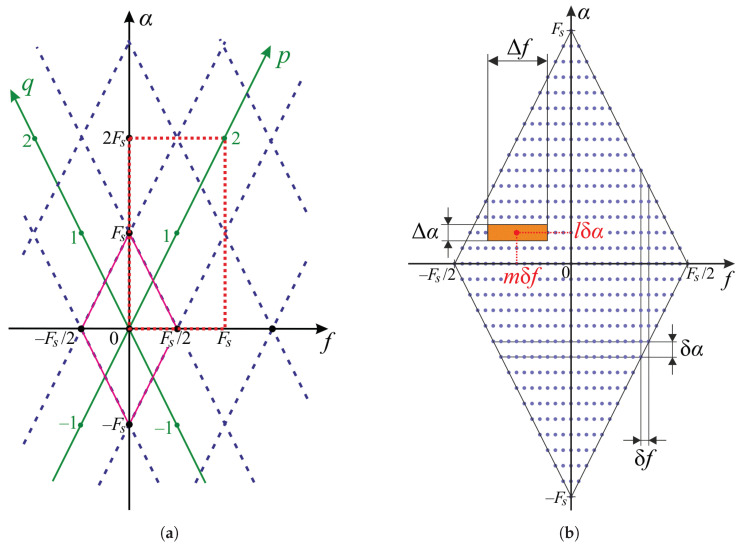
Structure of the “frequency–cyclic frequency” plane: general (**a**) Tiling of the “frequency–cyclic frequency” plane (The principal rhombus centered at the origin is drawn in pink) and detailed (**b**) SCF resolution cell in the bispectral plane and the covering grid consisting of the nodes where SCF is to be evaluated.

**Figure 2 sensors-23-00215-f002:**
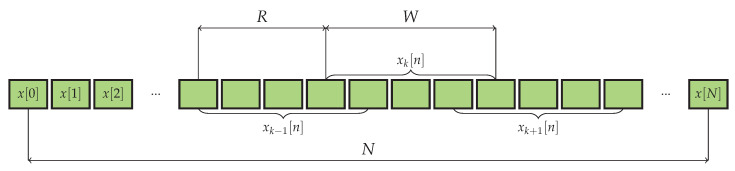
Sequence fragmentation.

**Figure 3 sensors-23-00215-f003:**
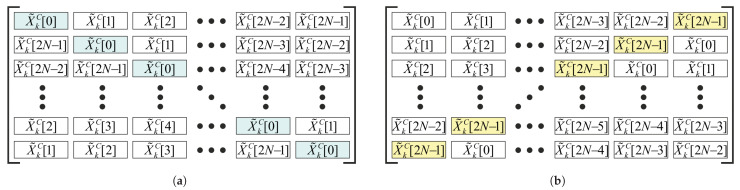
The structures of the matrices: (**a**) circulant matrix XR(k) and (**b**) anticirculant matrix XL(k).

**Figure 4 sensors-23-00215-f004:**
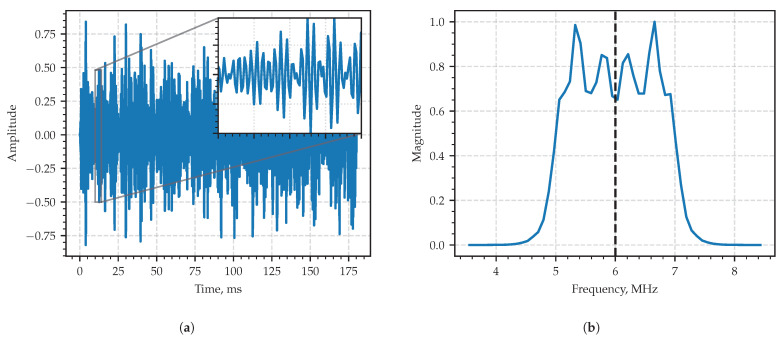
Fragment of the first example—amplitude-modulated signal x(t) in the time domain (**a**) and in the frequency domain (**b**). Dashed vertical line in the frequency domain corresponds to the value of the carrier frequency.

**Figure 5 sensors-23-00215-f005:**
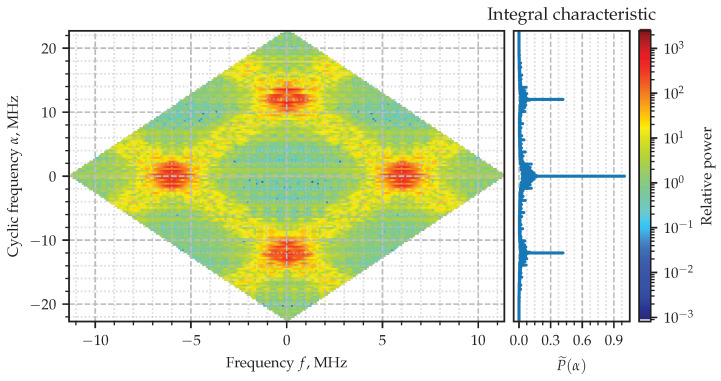
Estimated spectral correlation density of the AM signal.

**Figure 6 sensors-23-00215-f006:**
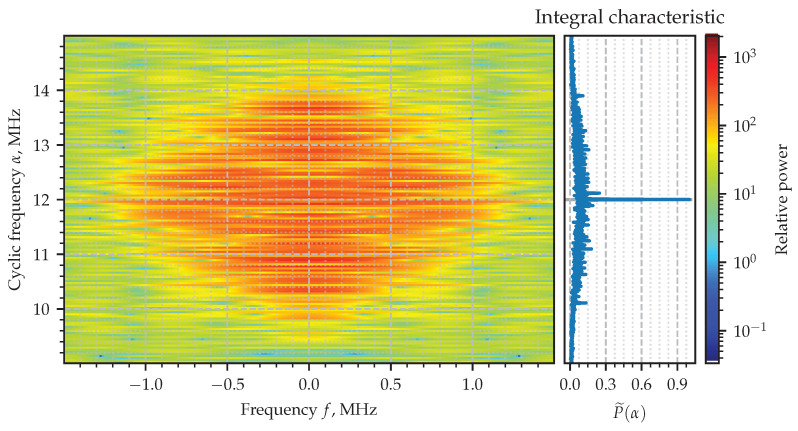
Estimated spectral correlation density of the AM signal.

**Figure 7 sensors-23-00215-f007:**
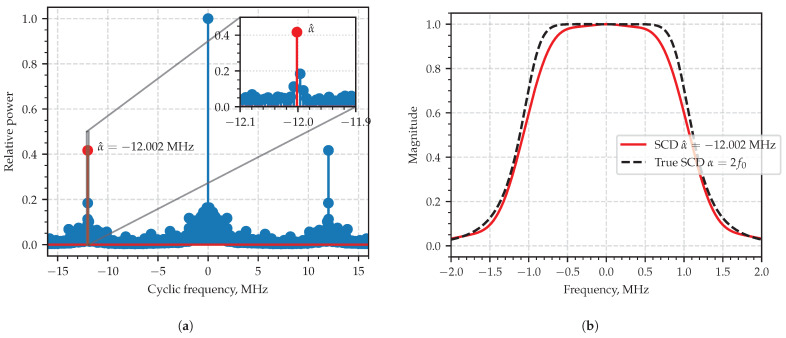
Area of interest in integral characteristics plot (**a**); the slice (absolute value) of the estimated SCF at the estimated characteristic cyclic frequency (**b**).

**Figure 8 sensors-23-00215-f008:**
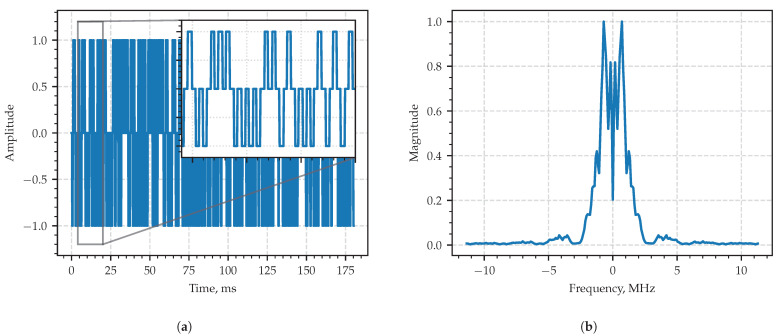
Fragment of the second example—PAM signal x(t) in the time domain (**a**) and in the frequency domain (**b**).

**Figure 9 sensors-23-00215-f009:**
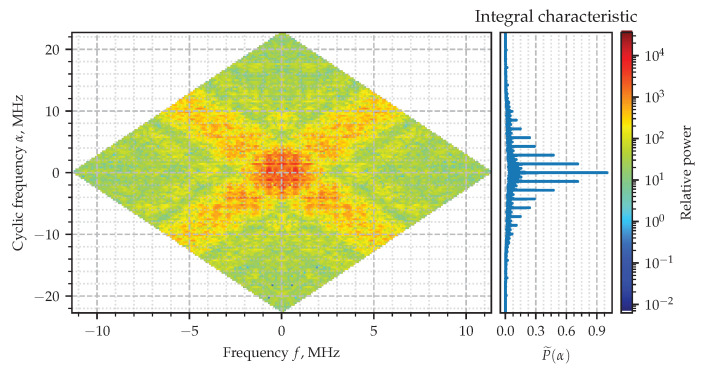
Estimated spectral correlation density of the PAM signal.

**Figure 10 sensors-23-00215-f010:**
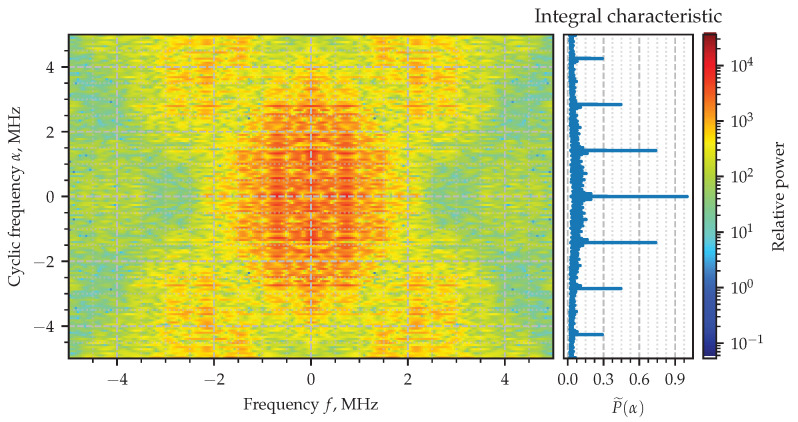
Estimated spectral correlation density of the PAM signal (zoomed version).

**Figure 11 sensors-23-00215-f011:**
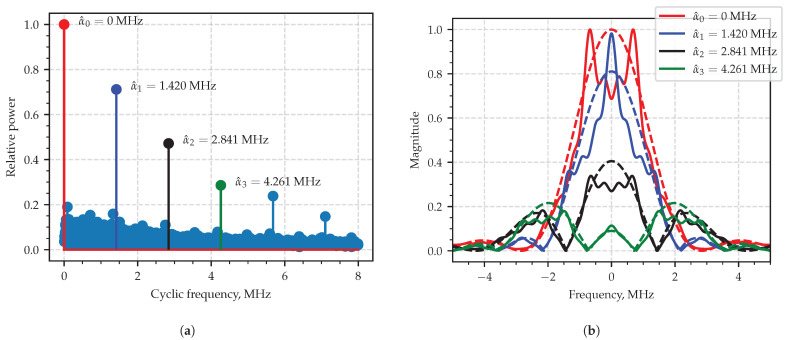
Area of interest in integral characteristics plot (**a**); SCF slice at estimated characteristic cyclic frequency (**b**): the theoretical curves are dashed lines, the estimated ones are solid.

**Table 1 sensors-23-00215-t001:** Memory cost and computational complexity of the main steps in the FAM algorithm.

Algorithm Step	Memory Cost	Computational Complexity
Sub-block matrix extraction	PW	
Data-tapering window application		PW
First Fourier transform		*P* times of FFT of size *W*
Phase correction		PW
Second Fourier transform	$PW^{2}$	$W^{3}$ multiplications plus FFT of size *P* times $W^{2}$
Total if *N* varies	O(N)	O(NlogN)
Total if *W* varies	$O(W^{2})$	$O(W^{3})$

**Table 2 sensors-23-00215-t002:** Memory cost and computational complexity of the main steps in the 2N-FFT algorithm.

Algorithm Step	Memory Cost	Computational Complexity
Initial buffer allocation	$4N^{2}$	
Data-tapering window application	*W*	PW
Fourier transform		*P* times of FFT of size 2N
Phase correction		PW
Cirulant and anti-circulant matrices processing	$3\times 4N^{2}$	$P\times 4N^{2}$ multiplications and additions
Total if *N* varies	$O(N^{2})$	$O(N^{3})$
Total if *W* varies	O(W)	O(W)

## Data Availability

Not applicable.

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
