# Peer review of "Estimation of a Spectral Correlation Function Using a Time-Smoothing Cyclic Periodogram and FFT Interpolation—2N-FFT Algorithm†"

_sensors, 2022, doi:10.3390/s23010215_

Round 1

Reviewer 1 Report

Suggestions: 

1-P.1, line 7: Write "double number fast Fourier transform (2N-FFT)" as said on page 17, line 491. 

2-P.1, line17: Replace the word “notional” with another with a more appropriate meaning. 

3-P.4, line 139: Write “spectral strip correlation algorithm (SSCA)”. 

4-P.10, lines 350/351: Write “fig.4a/fig.4b”. 

5-P.11, line 367: Identify in full the acronym corresponding to "PSD". 

6-P.12: Invert the positions of figures and text on the page. 

7-P.13, lines 386/387: Write “Fig.7a / fig.7b”. 

8-P.13, line 393: Replace “series” with “sequence”. y(t) is a series. 

9-P.15, line 420: Write “frequency accumulation method (FAM)”.

Author Response

In the attaced file you could find our response to your comments and the paper with all the changes we done during the revision.

Author Response

In the attached file you could find our response to your comments and the paper with all the changes we done during the revision.

Reviewer 3 Report

The paper presents a method for estimation of the spectral correlation function. From the presented results, the solution is properly working. My only concern is related to the amount of novelty w.r.t. reference [42], my suggestion is to explicitly write what is new here, i.e. PAM example and complexity computation, and something else? Finally I'd ask authors to rephrase to not replicate identical phrases from the previous work. Please check typos.

Author Response

(The authors gave the same response as above.)

Round 2

Reviewer 3 Report

The authors have properly replied to my previous comments.